# Efficacy of Combining an Extraoral High-Volume Evacuator with Preprocedural Mouth Rinsing in Reducing Aerosol Contamination Produced by Ultrasonic Scaling

**DOI:** 10.3390/ijerph19106048

**Published:** 2022-05-16

**Authors:** Shoji Takenaka, Maki Sotozono, Asaka Yashiro, Rui Saito, Niraya Kornsombut, Traithawit Naksagoon, Ryoko Nagata, Takako Ida, Naoki Edanami, Yuichiro Noiri

**Affiliations:** Division of Cariology, Operative Dentistry and Endodontics, Faculty of Dentistry & Graduate School of Medical and Dental Sciences, Niigata University, Niigata 951-8514, Japan; stakenaka@dent.niigata-u.ac.jp (S.T.); asaka846118@gmail.com (A.Y.); rsaito@dent.niigata-u.ac.jp (R.S.); nirayak@nu.ac.th (N.K.); naksagoon-ttw@dent.niigata-u.ac.jp (T.N.); lemmings@dent.niigata-u.ac.jp (R.N.); tida@dent.niigata-u.ac.jp (T.I.); edanami@dent.niigata-u.ac.jp (N.E.); noiri@dent.niigata-u.ac.jp (Y.N.)

**Keywords:** extraoral high-volume evacuator, preprocedural mouth rinsing, aerosols, povidone-iodine, essential oil, oral bacteria, dental office

## Abstract

The coronavirus disease pandemic has afforded dental professionals an opportunity to reconsider infection control during treatment. We investigated the efficacy of combining extraoral high-volume evacuators (eHVEs) with preprocedural mouth rinsing in reducing aerosol contamination by ultrasonic scalers. A double-masked, two-group, crossover randomized clinical trial was conducted over eight weeks. A total of 10 healthy subjects were divided into two groups; they received 0.5% povidone-iodine (PI), essential oil (EO), or water as preprocedural rinse. Aerosols produced during ultrasonic scaling were collected from the chest area (PC), dentist’s mask, dentist’s chest area (DC), bracket table, and assistant’s area. Bacterial contamination was assessed using colony counting and adenosine triphosphate assays. With the eHVE 10 cm away from the mouth, bacterial contamination by aerosols was negligible. With the eHVE 20 cm away, more dental aerosols containing bacteria were detected at the DC and PC. Mouth rinsing decreased viable bacterial count by 31–38% (PI) and 22–33% (EO), compared with no rinsing. The eHVE prevents bacterial contamination when close to the patient’s mouth. Preprocedural mouth rinsing can reduce bacterial contamination where the eHVE is positioned away from the mouth, depending on the procedure. Combining an eHVE with preprocedural mouth rinsing can reduce bacterial contamination in dental offices.

## 1. Introduction

Since 2020, the coronavirus disease (COVID-19) pandemic has affected many people all over the world [1]. Disease clusters, which are small-scale groups of infected persons, are meaningful with respect to investigations into the trends of COVID-19 infections [2]. Various sources of such infections have been reported, such as nursing homes [3], medical hospitals [4,5], families [6], schools [7], and restaurants [8]. Of these, hospital-acquired infections have become a serious problem [9]. Due to the spread of COVID-19, increased attention is being devoted to the importance of infection control in medical hospitals and clinics, not only regarding COVID-19, but also other related infectious diseases.

Many kinds of drug-resistant bacteria, such as the methicillin-resistant and vancomycin-resistant *Staphylococcus aureus*, are associated with hospital-acquired infections [10,11]. Preventing nosocomial infections is one of the most important tasks for hospital staff. In this regard, standard precautions refer to the minimum infection prevention practices that are recommended for all aspects of patient care in medical hospitals. These precautions consist of hand hygiene, the use of personal protective equipment, respiratory hygiene and coughing etiquette, sharps safety, safe injection practices, sterile instruments and devices, and clean and disinfected environmental surfaces [12].

Standard precautions for dental hospitals are also described in the United States of America Center for Disease Control guidelines; these precautions are currently widely implemented [13]. As dentists and dental hygienists use rotary cutting instruments and/or ultrasonic generators in the oral cavity, special attention should be paid to infections caused by splatters, droplets, and aerosols containing saliva and blood [14,15]. Therefore, in addition to the standard precautions adopted in medical hospitals, additional measures must be taken to prevent the spread of infectious diseases in dental clinics [16].

In previous studies, several methods and pieces of equipment have been suggested for minimizing aerosol production, including intra [17,18] or extra [18,19] oral high-volume evacuators (eHVEs), rubber dams [20], local stand-alone air cleaning systems [21], antimicrobial coolants [21], and preprocedural mouth rinsing [22,23]. Although eHVEs can reduce aerosol generation during ultrasonic scaling procedures by 90% or more [18,19], the positioning of an eHVE relative to the oral cavity is important for the effective mitigation of aerosols. However, different dental procedures may require the eHVE to be kept a certain distance away. Despite this, to date, the effect of the distance between the eHVE and oral cavity on the suppression of aerosol production remains unknown.

Preprocedural mouth rinsing has been reported to be effective in preventing contamination and infection in dental clinics, as it can reduce oral bacteria in aerosols and scattered materials that are caused by dental treatment since 1990s [16,24,25]. Due to the recent COVID-19 pandemic, the preprocedural mouthwash in dental clinics has been getting the attention of medical workers again. The human oral cavity is inhabited by more than 700 bacterial species; these bacteria form biofilms at different locations within the oral cavity [26,27,28]. Some clinical investigations have recently reported that using mouthwash, such as povidone-iodine (PI), chlorhexidine gluconate (CHG), cetylpyridinium chloride CPC), or essential oil (EO), can effectively reduce the viral load of not only oral bacteria but also of severe acute respiratory syndrome coronavirus 2 (SARS-CoV-2) in saliva [29,30,31]. Moreover, in a systematic review on preprocedural mouthwash, Garcia-Sanchez et al. reported that preprocedural mouthwash using PI is effective against SARS-CoV-2 in saliva and can decrease the risk of infections that may happen in dental treatments and interventions [32].

This study, thus, aimed to investigate the efficacy of combining an eHVE with preprocedural mouth rinsing in reducing contamination from aerosols produced by an ultrasonic scaler. In particular, this study focused on investigating the effect of the eHVE position on the suppression of aerosol generation.

## 2. Materials and Methods

### 2.1. Study Population

In total, 10 healthy subjects (four females and six males; age range: 26–51 years; mean age: 31.2 years) participated in this study. All the subjects who met the inclusion/exclusion criteria signed an informed consent form before participating in this study. The study protocol was approved by the Niigata University Ethics Committee (approval number: 2020-0113), and the methods were carried out in accordance with the approved guidelines.

### 2.2. Inclusion and Exclusion Criteria

The inclusion criteria were as follows: (a) a minimum of 20 natural teeth and (b) an O’Leary’s plaque control record of 20–30%. The exclusion criteria were as follows: (a) presence of orthodontic bands, (b) carious lesions requiring immediate restorative treatment, (c) subjects with a history of allergy to PI or EO (which were used in this study), (d) presence of one site with ≥5 mm probing pocket or presence of clinical attachment loss > 5 mm, (e) subjects who had taken local and/or systematic antibiotics within the previous 4 weeks, (f) subjects who had used antiseptic mouthwash in the last 4 weeks, (g) subjects who were smokers, (h) subjects who were pregnant or lactating, and (i) subjects who consumed excessive alcoholic beverages. In this study, excessive consumption of alcoholic beverages was defined as an intake of more than 20 g of pure alcohol in accordance with the guidelines of Ministry of Heath, Labour and Welfare in Japan [33].

### 2.3. Study Design

This study was designed as a double-masked, two-group, crossover randomized clinical trial; it was conducted over a period of 8 weeks (Figure 1). The types of mouthwash used in this study were 0.5% PI (Mundipharma K. K., Tokyo, Japan) and EO (Listerine fresh mint, Johnson & Johnson K. K., Tokyo, Japan). The effect of preprocedural mouthwash on the reduction of viable bacteria in oral aerosols was assessed after dental prophylaxis conducted with an ultrasonic scaler, using an eHVE (Free Arm ARTEO-T, Tokyo Giken, Inc., Tokyo, Japan). Subjects who did not undergo rinsing served as the control. The subjects were instructed to brush their teeth using the Bass brushing technique and to use the same kind of toothbrush and toothpaste during the experimental period.

A blood agar plate (tryptone soya agar [TSA] with 5% sheep blood agar; Becton Dickinson Company Ltd., Fukushima, Japan) and a Petri dish (85 mm in diameter) filled with 10 mL of sterilized saline were used to collect airborne microorganisms. Five standardized locations were evaluated for each treatment group. For each subject, one location was positioned at the patient’s chest area (PC), and the others being positioned at the dentist’s mask (DM), dentist’s chest area (DC), bracket table (BT), and assistant’s area (AA; Figure 2A,B). Each plate was fixed with adhesive tape. Petri dishes without distilled water were placed on the mask and chest. The Petri dish was fixed to the doctor’s mask using double-sided tape. The distance from the patient’s mouth to the agar plate is shown in Figure 2A,B. The effects of the eHVE (Figure 2C) were measured at distances 10 and 20 cm away from the mouth (Figure 2D).

The same operatory was used for each experiment; it was cleaned with 70% ethanol between uses. No regular patients were treated in this operatory during the study. The dental procedures (ultrasonic scaling and PMTC) were performed at a dental unit (GMP3-S, Osada Electric Co., Ltd., Tokyo, Japan) in a single room of Niigata University Medical and Dental Hospital. The air conditioner was always on and the room temperature was maintained at 22 °C during the procedure. The doors and windows were closed during the experiment. The volunteers received professional scaling and polishing (PMTC) to remove all calculus, plaque, and extrinsic tooth stains prior to the experiment. The first sampling started one week after the PMTC. The subjects had lunch at 12:00 and brushed their teeth after meal on the day when samples were collected. The experiments (ultrasonic scaling and PMTC) were performed at 17:00 (5 h after lunch). The subjects were instructed to refrain from eating and oral care, such as brushing, flossing, and using mouthwashes, during 5 h between brushing after lunch and the experiments. The subjects were randomly assigned to either the control (no rinsing; NR) or the test (rinsing with distilled water; DW) group using the closed envelope technique. The subjects allocated to the DW group rinsed for 30 s with 20 mL of solution. The volunteers were instructed not to inform their clinician as to whether or not they had rinsed with a given solution. Full-mouth scaling was conducted using a piezoelectric ultrasonic scaler (ST08, Osada Electric Co., Ltd., Tokyo, Japan) at a frequency of 25 kHz for 10 min; it was performed by a blinded clinician (one of the authors). All full-mouth dental prophylaxes in this study were performed by the same clinician; intraoral suction was performed during each treatment.

After each treatment, 10 mL of sterilized water was added to the empty Petri dish. The number of viable cells in the dish was determined by quantifying the amount of adenosine triphosphate (ATP) using the CellTiter-Glo 2.0 assay (Promega Corporation, Madison, WI, USA), according to the manufacturer’s instructions. Relative light units (RLUs) were measured using a microplate reader (GloMax Discover System GM3000, Promega Corporation). Colony-forming unit (CFU) counting was also performed to determine the number of viable cells. Agar plates were incubated anaerobically for 5 d.

A washout period of 7 d was instituted between each treatment following tooth polishing. Each subject was assigned to another group, and further data were collected. Following the abovementioned NR and DW experiments, subjects were randomly assigned to either a group that rinsed using 0.5% PI (PI group) or a group that rinsed using EO (EO group; Figure 1). In either case, subjects were instructed to rinse for 30 s with 20 mL of solution. After collecting data at a position where the distance between the eHVE and the mouth was 10 cm, data were collected using the same protocol with a distance of 20 cm.

### 2.4. Total Cell Counting

Quantitative analysis of the total bacteria in sterilized water, as detailed above, was performed using the polymerase chain reaction (PCR)-invader method (BML, Inc., Tokyo, Japan), as described previously [34].

### 2.5. Viable Cell Counting in the Mouth after Preprocedural Mouth Rinsing

To determine the reduction of viable cells in the mouth after preprocedural mouth rinsing, the subjects were asked to rinse with 20 mL of sterile saline for 30 s, followed by rinsing with 20 mL of DW, PI, or EO. Subjects who received the saline sample without rinsing with a solution served as control. The samples were homogenized, serially diluted, plated on TSA agar, and incubated anaerobically for 5 d at 37 °C. A washout period of 7 d was instituted between the sampling.

### 2.6. Statistical Analysis

Statistical analyses were performed using SPSS version 28 (IBM Corp., Armonk, NY, USA) and Excel Statistics 7.0 (Esumi Co., Ltd., Tokyo, Japan). Viable cell counts and ATP assays in dental aerosols were compared using the Friedman and Bonferroni tests. Viable cell counts in the mouth were compared with the Kruskal–Wallis test, and with a post hoc Steel–Dwass test. Statistical significance was set at *p* < 0.05.

## 3. Results

### 3.1. Baseline Demographic and Clinical Characteristics

The average number of teeth was 27.1 ± 1.7 and the average O’Leary’s plaque control record was 25.0 ± 3.5. All subjects remained in good health for the duration of the experiment, and each one participated in the entire series of experiments.

### 3.2. Viable Bacteria in Dental Aerosols

At a distance of 10 cm from the mouth, the eHVE successfully inhibited aerosol contamination produced by the ultrasonic scaler; viable bacteria amounted to 20 CFUs on average (without gargling), even on the chest of the patient with the highest degree of contamination (Figure 3A). No significant differences were observed among the different experimental groups or locations. The results of the ATP assay showed a similar trend, with little contamination among all groups, at all locations (Figure 4A, *p* > 0.05).

When the eHVE was positioned 20 cm away from the mouth, more dental aerosols containing bacteria were detected at the DC and PC. When the participants did not rinse their mouths, the number of CFUs at a distance of 20 cm increased 15-fold in the DC group and 9-fold in the PC group, compared with those at a distance of 10 cm (Figure 3B, Appendix A). The PC was the most contaminated site among all the studied locations. The DW group showed a slight reduction in the number of CFUs, but this difference was not significant. Both types of mouthwashes (PI and EO) showed significant reduction (up to 38% for PI and 33% for EO) at the PC, compared with the NR group. There was no significant difference between the effects of PI and EO.

The results of the ATP assay showed a similar tendency as those of CFU counting. The PC was the most contaminated site among all the studied locations (Figure 4B), with the other locations having similar RLUs. At a distance of 20 cm, RLUs increased 19-fold at the PC, compared with that at a distance of 10 cm. Rinsing with all kinds of solutions efficiently reduced the RLUs. At the PC, reductions of up to 62, 31, and 22% were obtained for water, PI, and EO, respectively, compared with the control (no rinsing). The RLUs of the PI and EO groups differed significantly from those of the control group (no rinsing).

### 3.3. Total Bacteria in Dental Aerosols

None of the samples could be quantitatively analyzed, indicating that the total number of bacteria in all the samples was below the detection limit of 3.7 log copies [35].

### 3.4. Viable Cell Counting in the Mouth after Preprocedural Mouth Rinsing

Figure 5 shows the viable bacteria counts in the mouth following the use of saline after rinsing with DW, PI, or EO. The participants rinsed with 20 mL of sterile saline for 30 s following the use of either solution. The NR group showed a viable count of 8.92 ± 0.7 log CFU. After rinsing with DW for 30 s, this decreased slightly, but not significantly. Rinsing with PI or EO reduced the viable cell count by 0.84 or 0.92 CFU, respectively; these decreases were significant compared with that in the control (no rinsing).

## 4. Discussion

The COVID-19 pandemic has allowed dental professionals to reconsider current practices regarding infection control during treatment. Here, focus was given to the effects of combining two infection control methods: eHVEs and preprocedural mouth rinsing. To the best of authors’ knowledge, this study represents the first investigation into the effects of the eHVE position and the adjunctive benefits of preprocedural mouth rinsing. Although previous studies have measured bacterial contamination by using CFUs as a proxy [21], here ATP assays were used in addition to CFU counting (Figure 3 and Figure 4). Some kinds of oral bacteria cannot grow on agar plates, which means that results obtained using CFU counting alone may not accurately reflect the infection status. Here, although the results of the CFU counting showed similar trends as those of the ATP assays, the latter was more sensitive, revealing that viable cells equivalent to those observed in the DM and DC groups were detected in the AA and BT groups (Figure 4B).

The eHVE, a popular device in Japan, differs from other high-volume evacuators described in some reports, and from those detailed in a systematic review on bio-aerosols in dentistry [36]. It has been reported that eHVEs can reduce the diffusion of aerosols throughout a dental office during ultrasonic scaling [19,37]. Here, when the eHVE was positioned 10 cm away from the subject’s mouth, the aerosol-generated bacterial contamination of the clinic was negligible (Figure 3 and Figure 4). This finding is consistent with a laboratory investigation performed by Horsophonphong et al. [37]. Thus, bringing the eHVE closer to the patient’s mouth can almost entirely prevent contamination.

When the eHVE was positioned 20 cm away from the mouth, bacterial contamination significantly increased at the DC and PC (Figure 3 and Figure 4). The PC was the most contaminated location measured in this study; the number of viable bacteria detected at the DC was more than nine times higher than that at the DC. These results indicate that reductions in aerosol and spatter depend on the distance between the eHVE and the mouth. Although the eHVE needs to be close to the patient’s oral cavity, aerosols can still be reduced to a certain extent if its distance from the oral cavity is large. Here, the total bacteria determined using PCR assays were under the detection limit of 3.7 log copies. Thus, eHVEs may be the best tool for protecting dental professionals, as emitted particles were attracted toward the PC, which was located near the eHVE.

Preprocedural mouth rinsing is a simple and cost-effective means of reducing the number of microorganisms in dental aerosols [38]. Furthermore, Chaudhary et al. demonstrated that mouth rinsing with 1% hydrogen peroxide, 0.12% chlorhexidine, or 0.5% PI for 60 s could decrease the viral load of SARS-CoV-2 in saliva by 61–89% at 15 min, and by 70–97% at 45 min [39]. The randomized-controlled clinical trial performed by Elzein et al. [40] revealed that mouthwash with 1% PI significantly reduced the salivary load of SARS-CoV-2, as evidenced by the delta Ct values, compared with the reduction achieved with distilled water (1% PI: 4.72 ± 0.89; distilled water: 0.519 ± 0.519). Recently, many studies on hospital-acquired infections focused on SARS-CoV-2 and there are few studies on bacterial contamination. In this study, mouth rinsing with either PI or EO for 30 s significantly reduced the number of bacteria in dental aerosols (Figure 3 and Figure 4) as well as in the oral cavity (Figure 5), when combined with an eHVE positioned 20 cm from the mouth. Although these studies investigated the viral load and it is difficult to compare with the result of this study on bacterial load, mouth rinsing with an antimicrobial agent provided an adjunctive benefit for infection control in the studied dental clinic. Paul et al. investigated the contamination of aerosols in ultrasonic scaling using methods similar to ours with three kinds of preprocedural mouthwashes (CHX, PI, and *Aloe vera*) [41]. Their results revealed that the number of bacteria at the patient’s chest area was higher than that at operator’s chest area for all the three mouthwashes. This result is similar to that obtained with an eHVE positioned 20 cm from the mouth (Figure 3B). When an eHVE was placed nearer to the mouth (10 cm), a reduction in CFU and RLU was observed (Figure 3A and Figure 4A). It is often difficult to perform a dental procedure while maintaining 10 cm distance between the patient’s mouth and eHVE in the clinical situation. Taken together, combining eHVEs and mouthwash could be recommended as a procedure for dental professionals.

However, the limitations of this study should be considered when interpreting these results. First, the sample size was small; only 10 subjects were recruited. In future studies, we would recruit more subjects and would consider the tendency depending on the attributes of subjects, such as age and sex. Moreover, these experiments were performed as a pilot study to investigate the combined effect of an eHVE and preprocedural mouth rinsing, as the experimental period lasted for eight weeks per subject. Furthermore, the oral cavities of the participants were relatively clean. Previous studies have included participants with relatively poor oral hygiene; for example, those with a mean plaque score of 1.8–3.0 [42], or those with >80% visible supragingival plaque on their tooth surfaces [43]. In this study, the identification of bacteria was not performed. Mouthwashes can affect the oral microbiome [44], and therefore, it is important to investigate the microbiome in aerosols produced during the dental procedure and to determine whether or not the microbiome changes depending on the mouthwash. The inhibitory effects of these techniques on dental aerosols may vary depending on the oral hygiene status of the patient.

## 5. Conclusions

Within the limitations of this study, combining an eHVE with mouth washing (using either 0.5% PI or EO) was found to reduce contamination from aerosols produced by an ultrasonic scaler. Although the eHVE was observed to prevent most bacterial contamination when positioned relatively close to the patient’s mouth, preprocedural mouth rinsing provided additional benefits in such situations where the eHVE must be positioned further away, depending on the dental procedure performed.

## Figures and Tables

**Figure 1 ijerph-19-06048-f001:**
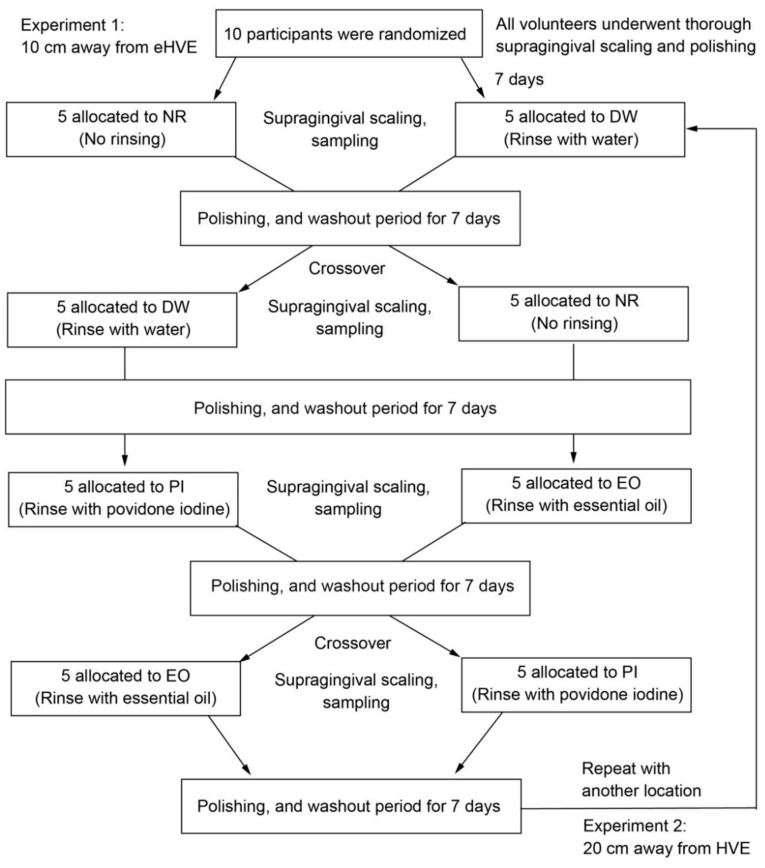
Flow chart of study design.

**Figure 2 ijerph-19-06048-f002:**
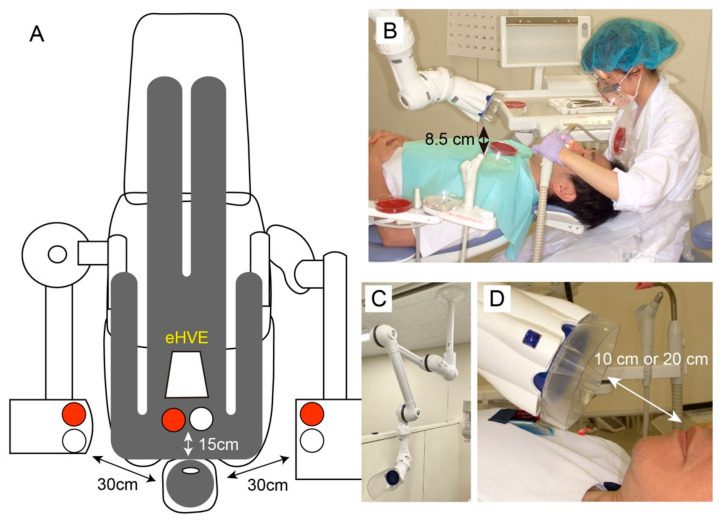
Positional relationship between sampling site and extraoral high volume evacuator (eHVE). (**A**) Dental unit and clinical test locations. Red and white circles indicate a blood agar plate and a Petri dish, respectively. (**B**) Positional relationship seen from side. (**C**) eHVE. (**D**) Positional relationship between eHVE and mouth. Distances of 10 and 20 cm were used.

**Figure 3 ijerph-19-06048-f003:**
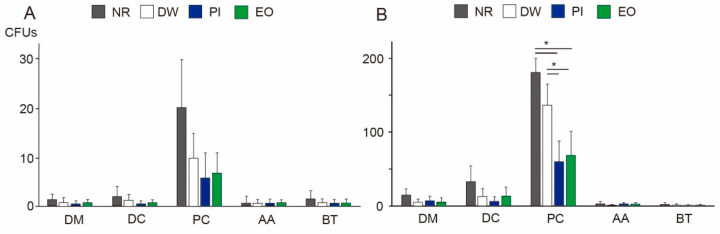
Viable counts on agar plates at various locations, following 10 min of scaling. Results show colony forming units (CFUs) at eHVE distances of (**A**) 10 cm and (**B**) 20 cm relative to the mouth. Results are shown as means ± standard deviation (SD). * *p* < 0.05. DM: doctor’s mask; DC: doctor’s chest area; PC: patient’s chest area; AA: assistant area; BT: bracket table; NR: no rinsing; DW: distilled water; PI: povidone-iodine; EO: essential oil.

**Figure 4 ijerph-19-06048-f004:**
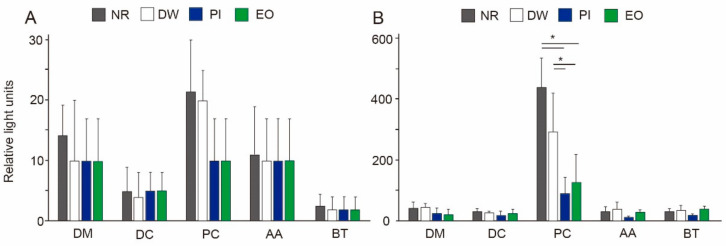
Relative light units (RLUs) at various locations following 10 min of scaling. Results show colony forming units (CFUs) at eHVE distances of (**A**) 10 cm and (**B**) 20 cm relative to the mouth. Results are shown as means ± SD. * *p* < 0.05.

**Figure 5 ijerph-19-06048-f005:**
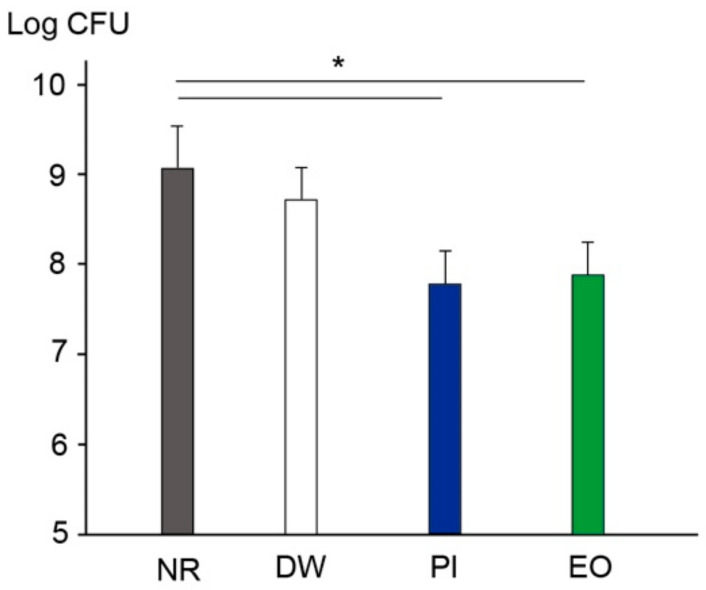
Viable counts in the mouth after rinsing with distilled water or mouthwash containing PI or EO. Results are shown as means ± SD. * *p* < 0.05.

## Data Availability

Data sets used during the study are available from the corresponding author on reasonable request.

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
