# Peer review of "Efficacy of Combining an Extraoral High-Volume Evacuator with Preprocedural Mouth Rinsing in Reducing Aerosol Contamination Produced by Ultrasonic Scaling"

_ijerph, 2022, doi:10.3390/ijerph19106048_

Round 1
Reviewer 1 Report
This article is new and innovative, and I congratulate its authors, but I would like to make some recommendations and ask some questions.
- How did you calculate the sample size?
- In the exclusion criteria, patients with excessive use of alcoholic beverages that may alter the bacterial flora.
- The age of the patients may have an influence, as it is a wide age range.
- How or which tool carried out the randomisation of each group.
- The results are somewhat small, they found differences between sexes for example.
- In the 8-week period the patient can change or modify their hygiene habit, performed measurements or motivation to maintain good hygiene.
- The discussion is somewhat narrow, there are articles measuring bacterial reduction with rinses, and some in reference to Covid-19.
Author Response
Responses to Comments from Reviewer 1
Comment 1: How did you calculate the sample size?
Response 1: We should have calculated the sample size statistically. However, we could not recruit more than 10 subjects because of the characteristics and personalities of patients who come to the Niigata University Medical and Dental Hospital.
We acknowledge that the small sample size is one of the limitations of the present study and have mentioned about it in the Discussion section, as described below.
Text: Lines 318–320
First, the sample size was small; only 10 subjects were recruited. In future studies, we would recruit more subjects and would consider the tendency depending on the attributes of subjects, such as age and sex.
Comment 2: In the exclusion criteria, patients with excessive use of alcoholic beverages that may alter the bacterial flora.
Response 2: As you have pointed out, the use of alcoholic beverages can change the oral microbiome. None of the patients recruited in this study used alcoholic beverages excessively. The criterion about excessive use of alcoholic beverages was added among the exclusion criteria. In this study, excessive use of alcoholic beverages was defined as an intake of more than 20 g of pure alcohol in accordance with the guidelines of Ministry of Heath, Labour and Welfare in Japan. [https://www.mhlw.go.jp/www1/topics/kenko21_11/b5.html]
The criterion has been added to the Materials and Methods section, as described below.
Revised text 2: lines 106–108
(i) subjects who used excessive alcoholic beverages. In this study, excessive use of alcoholic beverages was defined as an intake of more than 20 g of pure alcohol in accordance with the guidelines of Ministry of Heath, Labour and Welfare in Japan.
Comment 3: The age of the patients may have an influence, as it is a wide age range.
Response 3: The age of subjects may indeed affect the number of bacteria in the oral cavity. However, the number of subjects in this study was few and it was, therefore, difficult to investigate the influence of age. There was no significant difference in the results with regard to the sex of subjects. We recruited subjects having O’Leary’s plaque control record of 20–30% to ensure that the number of bacteria in the oral cavity of the subjects was equal to the extent possible (inclusion criteria (b), line 95).
In future studies, we would recruit more subjects and would consider the tendency depending on age and sex, because small sample size is one of the limitations of this study as described in the following sentences that have been added to the Discussion section.
Text: Lines 318–320
First, the sample size was small; only 10 subjects were recruited. In future studies, we would recruit more subjects and would consider the tendency depending on the attributes of subjects, such as age and sex.
Comment 4: How or which tool carried out the randomisation of each group.
Response 4: The subjects were randomly assigned to either the control (no rinsing; NR) or the test (rinsing with distilled water; DW) group using the closed envelope technique. This information has been added in the Materials and Methods section as mentioned below.
Revised text 4: lines 155–157
The subjects were randomly assigned to either the control (no rinsing; NR) or the test (rinsing with distilled water; DW) group using the closed envelope technique.
Comment 5: The results are somewhat small, they found differences between sexes for example.
Response 5: The age of the subjects may affect the number of bacteria in the oral cavity. However, the number of subjects in this study was small and it was difficult to investigate the influence of age. There was no significant different in the results of this study between the different sexes of subjects as described in our response to comment 3.
In future studies, we would recruit more subjects and would consider the tendency depending on the age and sex. The small sample size is one of the limitations of this study as we have mentioned in the Discussion section.
Revised text 3,5: Lines 318–320
First, the sample size was small; only 10 subjects were recruited. In future studies, we would recruit more subjects and would consider the tendency depending on the attributes of subjects, such as age and sex.
Comment 6: In the 8-week period the patient can change or modify their hygiene habit, performed measurements or motivation to maintain good hygiene.
Response 6: Although teeth brushing instructions were not given during the experimental period, patients recruited in this study maintained oral hygiene long time before the experiments. The subjects were instructed before the experiments to brush their teeth using the Bass brushing technique and to use the same kind of toothbrush and toothpaste during the experimental period. There was no significant difference in the plaque control record before and after experiments. We have mentioned about the oral hygiene habits of subjects in the Materials and Methods section as follows:
Revised text 6: Lines 118–120
The subjects were instructed to brush their teeth using the Bass brushing technique and to use the same kind of toothbrush and toothpaste during the experimental period.
Comment 7: The discussion is somewhat narrow, there are articles measuring bacterial reduction with rinses, and some in reference to Covid-19.
Response 7: We have added the following text to the Discussion section:
Revised text: Lines 296–314
The randomized-controlled clinical trial performed by Elzein et al. [40] revealed that mouthwash with 1% PI significantly reduced the salivary load of SARS-CoV-2, as evidenced by the delta Ct values, compared with reduction achieved with distilled water (1% PI: 4.72 ± 0.89; distilled water: 0.519 ± 0.519). Recently, many studies on hospital-acquired infections focused on SARS-CoV-2 and there are few studies on bacterial contamination. In this study, mouth rinsing with either PI or EO for 30 s significantly reduced the number of bacteria in dental aerosols (Figure 3 and 4) as well as in the oral cavity (Figure 5), when combined with an eHVE positioned 20 cm from the mouth. Although these studies investigated the viral load and it is difficult to compare with the results of this study on bacterial load, mouth rinsing with an antimicrobial agent provided an adjunctive benefit for infection control in the studied dental clinic. Paul et al. investigated the contamination of aerosols in ultrasonic scaling using methods similar to ours with three kinds of preprocedural mouthwashes (CHX, PI, and Aloe vera) [41]. Their results revealed that the number of bacteria at the patient’s chest area was higher than that at operator’s chest area for all the three mouthwashes. This result is similar to that obtained with an eHVE positioned 20 cm from the mouth (Figure 3B). When an eHVE was placed nearer to the mouth (10 cm), a reduction in CFU and RLU was observed (Figure 3A and 4A). It is often difficult to perform dental procedure keeping a 10 cm distance between the patient’s mouth and eHVE in the clinical situation.
Reviewer 2 Report
The research attempt was good but it has many methodological issues which raise concerns
- Page 2, line 79, what is inventions mean here?
- Why was chlorhexidine not used or considered?
- Why there were two controls - no rinsing and rinsing with distilled water
- How was the dish stabilized on the doctor mask??
- Inclusion criteria: O’Leary’s plaque control record of 20-30%. How can we have the same amount at all stages of the trial??
- Why were subjects with PD and CAL were excluded??
- Is it ethical for researchers to perform scaling and polishing for so many times for research??
- What is the rationale for this statement “The participants were instructed to refrain from using any other 137 means of oral hygiene and were asked to avoid eating or drinking 5 h before sampling” in fact if they eat, there might be more bacterial load
- How was log transformation done? Was the data still non-parametric after transformation?
- Page 5, line 189: At a distance of 10 m???
- Figure 3a: the scale on the y- axis has to be adjusted to appreciate the bars and values in the graph.
- Please describe more about the dental operatory.. was it a closed room with airconditioning or there is ventilation or is there in other equipment which can influence the air flow in the operatory?
- Please consider putting the results in tabular format also.
- How were the participants oral hygiene practices standardized??
- What are the microorganisms that were obtained in the CFU?? Was any attempt done to identify them??
- Page 7, line 233: with not significantly??
- How would the readers or reviewers know the facts mentioned? “When the partici-209 pants did not rinse their mouths, the number of CFUs at a distance of 20 cm increased 15-210 fold in the DC group and 9-fold in the PC group, compared to those at a distance of 10 cm”
- When the partici-209 pants did not rinse their mouths, the number of CFUs at a distance of 20 cm increased 15 fold in the DC group and 9-fold in the PC group, compared to those at a distance of 10 cm – Does this mean we don’t need pre-procedural rinse when we use eHVE? Then why do we need to combine eHVEs and mouthwash as recommended in discussion?
- What is the standard distance as recommended by manufacturer for placing the eHVE??
Author Response
Responses to Comments from Reviewer 2
Comment 1: Page 2, line 79, what is inventions mean here?
Response 1: We intended to write “intervention.” We are sorry for this typographical error.
Revised text: lines 79–82
Moreover, a systematic review regarding preprocedural mouthwash by Alvaro Gar-cia-Sanchez et al. reported that preprocedural mouthwash using povidone iodine is effective against SARS-CoV-2 in saliva and can decrease the risk of infections, which may happen in dental treatments and interventions [32].
Comment 2: Why was chlorhexidine not used or considered?
Response 2: Because allergic reactions to chlorhexidine pose problems, only mouthwashes containing less than 0.05% chlorhexidine are approved in Japan. We did not use chlorhexidine mouthwash because the concentration was different from that (0.2%) used worldwide.
Comment 3: Why there were two controls - no rinsing and rinsing with distilled water
Response 3: Before the COVID-19 pandemic, dentists and staff in dental offices and hospitals often had patients rinsing their mouth with water before dental treatment; we, therefore, kept rinsing water and no rinsing as controls. Moreover, in a previous study, preprocedural mouthwash with water tended to reduce CFU compared with that in the no-rinsing control [23]. We decided the control groups (no-rinsing and distilled water) in accordance with this study [23].
Comment 4: How was the dish stabilized on the doctor mask??
Response 4: The dish was fixed to the doctor’s mask using double-sided tape. This information has been added to the Materials and Methods section.
Revised text 4: Lines 132–133
Petri dishes without distilled water were placed on the mask and chest. The petri dish was fixed to the doctor’s mask using double-sided tape.
Comment 5: Inclusion criteria: O’Leary’s plaque control record of 20-30%. How can we have the same amount at all stages of the trial??
Response 5: Although teeth brushing instructions were not given during the experimental period, patients recruited in this study maintained oral hygiene long time before the experiments. The subjects were instructed before the experiments to brush their teeth using the Bass brushing technique and to use the same kind of toothbrush and toothpaste during the experimental period. There was no significant difference in the plaque control record before and after experiments. We have mentioned about the oral hygiene habits of subjects in the Materials and Methods section as follows:
Revised text 5: Lines 118–120
The subjects were instructed to brush their teeth using the Bass brushing technique and to use the same kind of toothbrush and toothpaste during the experimental period.
Comment 6: Why were subjects with PD and CAL were excluded??
Response 6: Patients with periodontitis and dental caries can have different oral microbiome and different number of bacteria in the oral cavity. Moreover, the results of CFU counting using blood agar plates can be dependent on the oral microbiome. Therefore, we excluded subjects with PD and CAL in this study.
Comment 7: Is it ethical for researchers to perform scaling and polishing for so many times for research??
Response 7: The study protocol of this study was approved by Niigata University Ethics Committee. Scaling and polishing are not invasive for subjects; there is no ethical problem in this study.
Comment 8: What is the rationale for this statement “The participants were instructed to refrain from using any other 137 means of oral hygiene and were asked to avoid eating or drinking 5 h before sampling” in fact if they eat, there might be more bacterial load?
Response 8: The description was confusing, and therefore, we have revised the experimental schedule as mentioned below. We considered that the bacterial load would be reduced before eating because dental biofilm on tooth surface is mechanically removed by mastication of food.
Revised text 8: Lines 151–155
The subjects had lunch at 12:00 and brushed their teeth after meal on the day when samples were collected. The experiments (ultrasonic scaling and PMTC) were performed at 17:00 (5 h after lunch). The subjects were instructed to refrain from eating and oral care, such as brushing, flossing, and using mouthwashes, during 5 h between brushing after lunch and the experiments.
Comment 9: How was log transformation done? Was the data still non-parametric after transformation?
Response 9: In accordance with a previous study, we did not perform log transformation for the data presented in Figures 4 and 5 [23]. The values of CFU were small for log transformation. Because the data were non-parametric, we used the Friedman and Kruskal–Wallis tests for statistical analysis.
Comment 10: Page 5, line 189: At a distance of 10 m???
Response 10: We apologize for this typographical error and have corrected it to 10 cm.
Revised text: Lines 208–211
At a distance of 10 cm from the mouth, the eHVE successfully inhibited aerosol contamination produced by the ultrasonic scaler; viable bacteria amounted to 20 CFUs on average (without gargling), even on the chest of the patient with the highest degree of contamination (Figure 3A).
Comment 11: Figure 3a: the scale on the y- axis has to be adjusted to appreciate the bars and values in the graph.
Response 11: The bars and values in Figure 3a were small in the original manuscript. We have corrected the y-axis and revised Figure 3 and Figure 4 (page 6).
Comment 12: Please describe more about the dental operatory.. was it a closed room with airconditioning or there is ventilation or is there in other equipment which can influence the air flow in the operatory?
Response 12: The dental procedures (ultrasonic scaling and PMTC) were performed at a dental unit (GMP3-S, Osada Electric Co Ltd, Tokyo, Japan) in a single room of Niigata University Medical and Dental Hospital. The air conditioner was always on and the room temperature was maintained at 22 °C during the procedure. The doors and windows were closed during the experiment. We have included this information in the Material and Methods section.
Revised text: Lines l44–149
The dental procedures (ultrasonic scaling and PMTC) were performed at a dental unit (GMP3-S, Osada Electric Co Ltd, Tokyo, Japan) in a single room of Niigata University Medical and Dental Hospital. The air conditioner was always on and the room temperature was maintained at 22 °C during the procedure. The doors and windows were closed during the experiment.
Comment 13: Please consider putting the results in tabular format also.
Response 13: Thank you for your advice. We have presented the values of CFU and RLU in a table as supplemental data.
Supplemental data
CFUs (20 cm)
|
|
DM |
DC |
PC |
AA |
BT |
|
No-rinsing |
14.7±8.3 |
32.6±21.6 |
180.9±18.7 |
3.2±3.6 |
2.1±2.1 |
|
DW |
5.1±5.2 |
12.4±15.4 |
136.2±29.3 |
1.0±1.5 |
1.4±1.6 |
|
PI |
6.9±8.9 |
6.6±5.4 |
59.8±27.9 |
2.4±3.6 |
0.6±1.1 |
|
EO |
5.7±5.2 |
13.4±13.8 |
68.6±32.2 |
2.3±2.8 |
1.3±1.6 |
CFUs (10 cm)
|
|
DM |
DC |
PC |
AA |
BT |
|
No-rinsing |
1.5±1.0 |
2.2±2.0 |
20.3±10.0 |
0.9±1.3 |
1.8±1.5 |
|
DW |
1.0±0.8 |
1.5±1.0 |
10.0±5.0 |
0.9±0.5 |
1.0±0.7 |
|
PI |
0.7±0.5 |
0.7±0.5 |
6.0±5.0 |
0.9±0.5 |
0.9±0.7 |
|
EO |
1.0±0.4 |
1.0±0.4 |
7.0±4.0 |
0.9±0.4 |
0.9±0.6 |
RLUs (20 cm)
|
|
DM |
DC |
PC |
AA |
BT |
|
No-rinsing |
41.0±18.9 |
30.8±9.4 |
408.7±127 |
31.4±14.8 |
30.7±9.7 |
|
DW |
42.4±14.8 |
26.6±5.8 |
293.1±127 |
39.6±22 |
33.7±16.0 |
|
PI |
24.2±16.7 |
18±13.9 |
88.4±54.5 |
10.5±4.9 |
18.5±3.5 |
|
EO |
20.7±17.7 |
23.2±14.8 |
126.8±90.5 |
27.3±8.0 |
38.4±10.7 |
RLUs (10 cm)
|
|
DM |
DC |
PC |
AA |
BT |
|
No-rinsing |
14.2±5.0 |
4.9±4.0 |
21.4±10.0 |
11.0±8.0 |
2.5±2.0 |
|
DW |
10.0±10.0 |
4.0±4.0 |
20.0±5.0 |
10.0±7.0 |
2.0±2.0 |
|
PI |
10.0±7.0 |
5.0±3.0 |
10.0±7.0 |
10.0±7.0 |
2.0±2.0 |
|
EO |
10.0±7.0 |
5.0±3.0 |
10.0±7.0 |
10.0±7.0 |
2.0±2.0 |
The results of CFU counts and ATP assay are shown in Table. The values in the parentheses are the distances between eHVE and the mouth.
DM: doctor’s mask; DC: doctor’s chest area; PC: patient’s chest area; AA: assistant’s area; BT: bracket table; NR: no rinsing; DW: distilled water; PI: povidone iodine; EO: essential oil.
Comment 14: How were the participants oral hygiene practices standardized??
Response 14: The subjects were instructed before the experiments to brush their teeth using the Bass brushing technique and to use the same kind of toothbrush and toothpaste during the experimental period. There was no significant difference in the plaque control record before and after experiments. We have mentioned about the oral hygiene habits of subjects in the Materials and Methods section as follows:
Revised text 6 and 14: Lines 118–120
The subjects were instructed to brush their teeth using the Bass brushing technique and to use the same kind of toothbrush and toothpaste during the experimental period.
- What are the microorganisms that were obtained in the CFU?? Was any attempt done to identify them??
Response 15: The detected bacteria were not identified because, in this study, we focused only on the number of bacteria in the aerosol. In future studies, it may be necessary to investigate the microbiome in aerosols produced during the dental procedure and to determine whether or not the microbiome changes depending on the kinds of mouthwash. We have described this limitation in the Discussion section as follows:
Revised text 15: Lines 326–329
In this study, the identification of bacteria was not performed. Mouthwashes can affect the oral microbiome [44], and therefore, it is important to investigate the microbiome in aerosols produced during the dental procedure and to determine whether or not the microbiome changes depending on the mouthwash.
Comment 16: Page 7, line 233: with not significantly??
Response 16: PI and EO reduced significantly compared with the no-rinsing. There was no significant difference between DW and no-rinsing, although RLUs tended to be lower in DW than in no-rinsing.
Comment 17: How would the readers or reviewers know the facts mentioned? “When the partici-209 pants did not rinse their mouths, the number of CFUs at a distance of 20 cm increased 15-210 fold in the DC group and 9-fold in the PC group, compared to those at a distance of 10 cm”
Response 17: The values of CFU and RLU have been added in the in table presented as supplemental data. The Results section was modified as mentioned below.
Revised text 17: Lines 228–232
When the participants did not rinse their mouths, the number of CFUs at a distance of 20 cm increased 15-fold in the DC group and 9-fold in the PC group, compared with those at a distance of 10 cm (Figure 3B, Supplemental data).
Comment 18: When the participants did not rinse their mouths, the number of CFUs at a distance of 20 cm increased 15 fold in the DC group and 9-fold in the PC group, compared to those at a distance of 10 cm – Does this mean we don’t need pre-procedural rinse when we use eHVE? Then why do we need to combine eHVEs and mouthwash as recommended in discussion?
Response 18: The results described in Figure 3 show that the eHVE has strong effect in reducing the CFU values. However, it is often difficult to perform a dental procedure with a distance of 10 cm between patient’s mouth and eHVE under clinical situations. Under clinical situations, the distance between patient’s oral cavity and eHVE is often more than 20 cm because of the positions. Therefore, we need to combine eHVEs and mouthwash. We have mentioned these points in the Discussion section as follows:
Revised text 18: Lines 307–314
Paul et al. investigated the contamination of aerosols in ultrasonic scaling using methods similar to ours with three kinds of preprocedural mouthwashes (CHX, PI, and Aloe vera) [41]. Their results revealed that the number of bacteria in the patient’s chest area was higher than that at operator’s chest area for all the three mouthwashes. This result is similar to that obtained with an eHVE positioned 20 cm away from the mouth (Figure 3B). When an eHVE was placed nearer to the mouth (10 cm), a reduction in CFU and RLU was observed (Figure 3A and 4A). It is often difficult to perform dental procedure keeping a 10 cm distance between patient’s mouth and eHVE in the clinical situation.
Comment 19: What is the standard distance as recommended by manufacturer for placing the eHVE??
Response 19: The standard distance was not prescribed by the manufacturer. It is considered that placing the eHVE as near to patient’s mouth as possible is effective.
Round 2
Reviewer 2 Report
All my queries were addressed.